Close neighbors, not intruders: investigating the role of tank bromeliads in shaping faunal microbiomes

Martínez-Mota Rodolfo 1 rodomartinez@uv.mx
Vásquez-Aguilar Antonio Acini 2
Hernández-Rodríguez Dolores 3
Suárez-Domínguez Emilio A. 4 emisuarez@uv.mx
http://orcid.org/0000-0002-1398-8172 Krömer Thorsten 1
1 Centro de Investigaciones Tropicales, Universidad Veracruzana , Xalapa, Veracruz , Mexico
2 Red de Biología Evolutiva, Instituto de Ecología, A.C. (INECOL) , Xalapa, Veracruz , Mexico
3 Instituto de Ciencias Básicas, Universidad Veracruzana , Xalapa, Veracruz , Mexico
4 Facultad de Biología and Museo de Zoología, Universidad Veracruzana , Xalapa, Veracruz , Mexico
Corte Guilherme
Electronic publication date: 2025 May 9
Publication date: 2025
Volume: 13
Electronic Location ID: e19376
Received 2024 Aug 26; Accepted 2025 Apr 7
Copyright: © 2025 Martínez-Mota et al.
Copyright year: 2025
Copyright holder: Martínez-Mota et al.
License: This is an open access article distributed under the terms of the Creative Commons Attribution License, which permits unrestricted use, distribution, reproduction and adaptation in any medium and for any purpose provided that it is properly attributed. For attribution, the original author(s), title, publication source (PeerJ) and either DOI or URL of the article must be cited.
License URL: https://creativecommons.org/licenses/by/4.0/

Keywords: Microbiome, Epiphytes, Tillandsia, Microbial transfer, Tropical cloud forest, Mexico, Diptera, Microhabitats

Funding: Consejo Nacional de Humanidades, Ciencias y Tecnologías (CONAHCYT) of Mexico This work was supported by Consejo Nacional de Humanidades, Ciencias y Tecnologías (CONAHCYT) of Mexico, under the scientist repatriation program 2019. The funders had no role in study design, data collection and analysis, decision to publish, or preparation of the manuscript.

==============================
Background

Tropical montane cloud forests contain high levels of epiphyte diversity. Epiphytic tank bromeliads play an important role in the functioning of these ecosystems and provide a microhabitat for many species of invertebrates. Microbial ecology theory suggests that the environment serves as a source of microbes for animals, but the contribution of this factor to the composition of an animal microbiome varies. In this study, we examined the extent to which tank bromeliads (Tillandsia multicaulis) serve as a source of microbes for two species of fly larvae in a cloud forest fragment in central Veracruz, Mexico.

Methods

We used 16S rRNA sequencing to characterize the bacterial communities in the organic matter within bromeliad tanks and in the whole bodies (surface and gut) of larvae from two fly taxa (Austrophorocera sp., Tachinidae, and Copestylum sp., Syrphidae) that inhabit these bromeliads. To assess the contribution of bromeliads to the microbiome of the fly larvae, we conducted fast expectation-maximization microbial source tracking (FEAST) analysis.

Results

The bacterial communities in bromeliad tanks were primarily composed of Pseudomonadota, Acidobacteriota, Bacteroidota, Verrucomicrobiota, and Spirochaetota. Similarly, communities of the fly larvae contained Pseudomonadota, Bacteroidota, Bacillota, and Actinomycetota. Bromeliad tanks exhibited the highest bacterial richness, followed by Copestylum and Austrophorocera larvae. Beta diversity analyses indicated that bacterial communities clustered by species. We found a modest contribution of bromeliads to the fly microbiome, with nearly 30% of the larvae microbiome traced to the organic matter deposited in the tanks.

Conclusions

Our data suggest that the microbiome of flies, which inhabit tank bromeliads during their larval stage, is nourished to some extent by the bacterial communities present in the organic matter within the tank.

Introduction

Community ecology theory suggests that species assemblages are influenced by interactions between biotic and abiotic factors, which act as ecological filters (Kraft et al., 2015). These intricate relationships may exert varying pressures on host-microbe symbiosis, leading to unique microbial community compositions and structures among different species sharing the same habitat. In particular, the animal microbiome is shaped by factors such as host phylogeny or environmental exposure (Douglas, 2018; Kohl, Dearing & Bordenstein, 2018; Weinstein et al., 2021), but their relative contributions can differ across species. Understanding these variations is essential for elucidating how the environment influences microbial communities across different hosts.

Microbial communities play a significant role in supporting the health of animals and plants, as well as in ecosystems services (Douglas, 2015; Saccá et al., 2017; Zhu et al., 2023; Compant et al., 2024). These communities are referred as the microbiome (Berg et al., 2020), and can be found in different environments including soil, decaying organic matter, the body surface of plants, and the gut of animals (Douglas, 2018; Banerjee & Van Der Heijden, 2023). Several studies across different biological systems have shown that ecological disturbance may disrupt host-microbiome ecological interactions, impacting diversity and ecological functions of microbial communities (Neely et al., 2022; Kiesewetter, Otano & Afkhami, 2023; Zhu et al., 2023). This calls for increasing efforts to understand the ecological drivers shaping microbial assembly in plants and animals inhabiting degraded ecosystems (Trevelline et al., 2019).

Tropical montane cloud forests are among the most biodiverse ecosystems in the Neotropics and Mexico (Doumenge et al., 2012; Gual-Díaz & Rendón-Correa, 2017), yet they are threatened by deforestation, habitat degradation, and climate change (Scatena et al., 2010; Sánchez-Ramos & Dirzo, 2014; López-Arce et al., 2019). Vascular epiphytes are an essential and integral component of these forests, characterized by high levels of plant diversity and endemism (Gentry & Dodson, 1987; Espejo-Serna et al., 2021; Taylor et al., 2022). These plants account for up to 50% of the plant species richness in cloud forests (Kelly et al., 1994).

Epiphytic tank bromeliads (Bromeliaceae) are particularly important, fulfilling key ecosystem services (Gotsch, Nadkarni & Amici, 2016; Ladino et al., 2019). Their water-retaining foliage (phytotelmata) creates microhabitats for canopy fauna offering nutrients, shelter, and suitable environmental conditions, which help sustain the ecological interactions among diverse assemblages of invertebrate and vertebrate species (Balke et al., 2008; Ospina-Bautista et al., 2008). Previous research has investigated the metabolic functions of microbial communities within phytotelmata (Gonçalves et al., 2014; Louca et al., 2017), with additional studies examining host-microbe resource competition in tank bromeliads (Rogy & Srivastava, 2023). However, the extent to which the organic matter deposited in tank bromeliads serves as a microbial pool that nourishes the microbiome of canopy fauna in the tropical cloud forest still remains an open question. To fill this gap, we employed microbial community profiling to explore the role of epiphytic tank bromeliads in sustaining symbiotic relationships of flies inhabiting tropical cloud forests.

We used fly larvae (Diptera) of two taxa as a model to test whether the tanks of Tillandsia multicaulis function as an environmental microbial source for the microbiome of flies inhabiting epiphytic bromeliads. Several insects take advantage of decaying organic matter accumulated in stagnant water for egg laying and larval development, especially in bromeliad phytotelmata (Batzer & Wissinger, 1996; Antonetti, Malfatti & Pinto-Utz, 2021; Lopes Filho et al., 2023). We predicted that if bromeliad tanks function as a microbial pool for fly larvae, then, a transfer of microbes from the organic material into the fly microbiome would occur.

Materials and Methods

Study site and sample collection

During December 2020, we conducted fieldwork in a fragment of disturbed cloud forest located at about 1,650 m a.s.l. in the central part of the State of Veracruz, at the Municipality of Tlalnelhuayocan, Mexico (19°32′47.1″N, 96°58′30.73″W; datum WGS84; Susan-Tepetlan, Velázquez-Rosas & Krömer, 2015). We accessed the canopy using the single-rope technique, and collected bromeliad individuals (n = 15) of the abundant epiphytic species Tillandsia multicaulis from the canopy of six oak trees (Quercus sp.), approximately 10–15 m above the ground. We placed tank bromeliads in plastic bags and transferred them to the laboratory at the Centro de Investigaciones Tropicales (Tropical Research Center) of Universidad Veracruzana, Mexico. We then collected the organic matter, consisting of wet soil and debris, from the tank of each plant. We examined the tank for fly larvae and collected individuals of two Diptera taxa (Austrophorocera sp., Tachinidae, and Copestylum sp., Syrphidae). We selected these fly taxa, which were consistently present in all collected bromeliads, to obtain sufficient replicates and ensure high confidence in our results. We preserved specimens in 70% ethanol, and stored samples at −20 °C until DNA extraction.

DNA extraction, microbial sequencing and processing

We extracted DNA from the organic matter of tanks (from here on “bromeliad tanks”) and larvae using a commercial DNA extraction kit (DNEasy PowerSoil QIAGEN), following the manufacturer’s instructions. For each fly larva, we extracted the DNA by crushing the entire larva (including the body surface and gut) within the lysis extraction tube. We sent the extracted DNA to the Integrated Microbiome Resource at Dalhousie University, Halifax, NS, Canada. The V6–V8 hypervariable region of the bacterial 16S rRNA gene was amplified using the primers B969F-BA1406R (Comeau et al., 2011). Sequencing was conducted on the Illumina MiSeq platform to produce 2 × 300 bp paired-end reads. The raw sequences were submitted to the NCBI Sequence Read Archive (SRA) under the submission number SUB14659115 (BioProject PRJNA1146765).

We processed bacterial sequences in the Quantitative Insights into Microbial Ecology 2 (QIIME2) bioinformatics platform, version qiime2-2020.2 (Estaki et al., 2020). We first removed primers using the q2-cutadapt plugin. Later, double-end sequences were subjected to quality control including denoising, merging, and chimera removal using the DADA2 pipeline (Callahan et al., 2016). We assigned bacterial reads as Amplicon Sequence Variants (ASVs) using the q2-dada2 plugin (Callahan, McMurdie & Holmes, 2017). We performed sequence taxonomic assignment using the naive Bayes classifier trained on the SILVA 132 99% OTUs sequence reference (Quast et al., 2013). We also performed filtering of chloroplast, mitochondria, doubletons, and removed ASVs present in less than 10% of our samples. Given the different nature of biological samples (organic matter from tanks and larvae) we processed them separately. After filtering, we obtained 850,675 reads for bromeliad tanks, with a median of 40,771 reads per sample, resulting in 2,390 ASVs. We ended up with 570,509 reads for Diptera after filtering, with a median of 16,492 reads per sample; this resulted in 438 ASVs. Afterwards, we merged the two feature tables in QIIME2. Because inherent differences in bacterial biomass and sequenced reads from bromeliad tanks and fly larvae, we rarefied samples to a minimum of 2,400 reads prior to downstream analyses. This helped to retain all of our samples, avoid losing information, and made microbial analysis from different sources more comparable. According to Caporaso et al. (2012), a minimum of 1,000 reads is enough to characterize bacterial communities; therefore, we are confident that reliable conclusions are drawn from these analyses.

Fly larvae identification

We achieved taxonomic identification of fly larvae through cytochrome c oxidase subunit I (COI) metabarcoding, using the LCO1490–CO1-CFMRa primers (Jusino et al., 2019). We processed these samples in the QIIME2 platform. The COI sequences obtained for each individual were verified in the BOLD System database (Ratnasingham & Hebert, 2007). To ensure that each individual belongs to the same taxonomic group, we applied a similarity threshold of 100% for taxonomic assignments.

Data analyses

We assessed changes in microbial alpha diversity by calculating the observed ASVs, which is an index of microbial richness. To gain a deeper understanding of the relationships among microbes, we also computed Faith’s phylogenetic diversity, which considers the total length of the phylogenetic tree branches within a community (Estaki et al., 2020). We calculated these metrics in QIIME2 and compared them among the three host species (i.e., T. multicaulis and the two fly taxa identified through COI metabarcoding) using Kruskal-Wallis tests. We then tested for the effects of host species on microbial beta diversity using PERMANOVAs. We used as response variables unweighted and weighted UniFrac distance matrices, which consider the presence and relative abundance of microbes, respectively, together with phylogenetic information (Lozupone et al., 2011). Given that differences in microbial communities can be influenced by dispersion within a group, we assessed within-community dispersion with the betadisper function of the vegan package (Oksanen et al., 2022). We depicted community shifts among species using Principal coordinate analysis (PCoA). We further explored microbial differential abundance among species with linear discriminant analysis effect size (LEfSe; Segata et al., 2011) using the microbiomeMarker package (Cao et al., 2022). We normalized microbial counts by the counts per million (CPM) method and set the lda score cutoff at 2; we adjusted p-values for false-discovery rate. Unless stated otherwise, we conducted all analyses in RStudio (R Core Team, 2024) using several packages, including phyloseq (McMurdie & Holmes, 2013), qiimer (Bittinger, 2015), qiime2R (Bisanz, 2018), and ggplot2 (Wickham, 2016).

Tracking the source of microbes

To determine the contribution of bromeliad tanks as a source of microbes for Diptera larvae, we conducted fast expectation-maximization microbial source tracking (FEAST) analysis (Shenhav et al., 2019). FEAST consists in estimating the proportion accounted by a particular microbial source sample into another microbiome recipient. We set T. multicaulis as a source of microbial communities for the two fly taxa (sink). We implemented this analysis in RStudio following the procedure found in https://github.com/cozygene/FEAST and detailed elsewhere (Shenhav et al., 2019). We also calculated the number of shared ASVs between bromeliads and the two Diptera larvae using the function ps_venn from the MicEco package (Russel, 2023).

Results

Microbial diversity between species

Using COI metabarcoding, we identified two fly taxa inhabiting bromeliad tanks: Austrophorocera sp. (Tachinidae, hereafter referred to as Austrophorocera) and Copestylum sp. (Syrphidae, hereafter referred to as Copestylum). Measures of microbial alpha diversity varied between bromeliad tanks and the two fly taxa. Significant differences were observed in both the number of ASVs and phylogenetic diversity (Kruskal-Wallis test: H = 30.7, p < 0.001). Bromeliad tanks exhibited the highest levels of microbial diversity compared to the two fly taxa (Kruskal-Wallis pairwise test: H = 18.3, p < 0.001; Figs. 1A and 1B). When comparing the two insect taxa, significant differences were observed, with Copestylum exhibiting two-fold and three-fold higher observed ASVs and phylogenetic diversity, respectively, than Austrophorocera (H = 15.0, p < 0.001).

Figure 1 Bacterial diversity of epiphytic bromeliad tanks and two fly larvae genera from the tropical montane cloud forest.

Data from bromeliads are depicted in green and data from flies are depicted in purple (Austrophorocera sp.) and dark golden (Copestylum sp.). (A, B) Bromeliads had higher alpha diversity than the two fly larvae. (C, D) Different bacterial community composition and structure were found among bromeliads and the two fly larvae measured by unweighted and weighted UniFrac distance matrices.

We also found distinct microbial communities among bromeliad tanks and the two fly taxa, as measured by unweighted and weighted UniFrac distance matrices (PERMANOVA: unweighted UniFrac, pseudo-F2,35 = 23.9, p < 0.001, R2 = 0.59; weighted UniFrac, pseudo-F2,35 = 9.8, p < 0.001, R2 = 0.37). PCoA plots showed that microbial communities clearly clustered by species, even between the two Diptera taxa. This pattern was more evident when using a metric that accounts only for microbial presence, i.e., unweighted UniFrac (Fig. 1C). The grouping patterns were not affected by within-community dispersion (betadisper: F2,33 = 0.7, p = 0.47, Fig. S1). However, when the relative abundance of microbes was considered (i.e., weighted UniFrac), within-community dispersion was detected (betadisper: F2,33= 17.9, p < 0.001, Fig. S1), most likely driven by Austrophorocera fly larvae (Fig. 1D).

Differential abundance among species

Each species showed a particular composition of microbes at distinct taxonomic levels (Fig. 2). At phylum level, bromeliad tanks showed a significant enrichment of Acidobacteriota (lda score = 5.2, fdr p-value < 0.001), Verrucomicrobiota (lda score = 4.7, fdr p-value < 0.001), Spirochaetota (lda score = 4.1, fdr p-value < 0.001), and Patescibacteria (lda score = 3.6, fdr p-value < 0.001), which were some of the most abundant microbial phyla in these plants. Less abundant taxa also were enriched (for a complete list see Table S1). In the case of the two fly larva species, Austrophorocera showed significant increases in Pseudomonadota (lda score = 4.8, fdr p-value < 0.05), Bacteroidota (lda score = 4.8, fdr p-value < 0.01), and Actinomycetota (lda score = 4.7, fdr p-value < 0.01). Copestylum was enriched in Bacillota (lda score = 5.2, fdr p-value < 0.001), Epsilonbacteraeota (lda score = 3.5, fdr p-value < 0.001), and Tenericutes (lda score = 3.5, fdr p-value < 0.001).

Figure 2 Relative abundance of bacterial phyla of epiphytic bromeliad tanks and two fly larvae genera from the tropical cloud forest in central Veracruz, Mexico.

Each bar represents an individual. Abundance is expressed as proportion (y-axis).

The relative abundance of microbial families also differed between species. Acidobacteriaceae (13.0%), Pedosphaeraceae (7.5%), and Acetobacteraceae (6.2%) showed the highest relative abundance in bromeliad tanks (Fig. 3). The families Chitinophagaceae and Beijerinckiaceae showed the highest abundance in the two fly taxa (20.3% and 12.2%, respectively, in Austrophorocera; 26.6% and 24.8%, respectively, in Copestylum). These bacteria were also found in the tanks of bromeliads (7.0% and 6.9%, respectively).

Figure 3 Relative abundance of microbial families found in epiphytic bromeliad tanks and in two fly larvae from the tropical montane cloud forest of central Veracruz, Mexico.

Abundance is expressed as proportion. The x-axis shows samples which are colored by species (purple: Austrophorocera sp.; dark golden: Copestylum sp.; green: bromeliad tanks).

In addition, several bacterial families were distinctly enriched in bromeliad tanks and fly larvae. Bromeliad tanks had a total of 128 enriched taxa, Austrophorocera a total of 21, and Copestylum a total of 27 enriched taxa. Bromeliads in particular, had significant enrichment of bacteria belonging to the WD2101 soil group (lda score = 4.0, fdr p-value < 0.001) and Xanthobacteraceae (lda score = 3.9, fdr p-value < 0.001). On the other hand, Austrophorocera larvae were enriched in Rhizobiales (lda score = 4.2, fdr p-value < 0.001) and Desulfovibrionaceae (lda score = 4.4, fdr p-value < 0.001), among others; and Copestylum were enriched in Bacteroidaceae (lda score = 4.1, fdr p-value < 0.001) and Clostridiales vadin BB60 group (lda score = 3.9, fdr p-value < 0.001). Table S1 showed a complete list of enriched bacterial families.

Contributions of bromeliad tanks to the fly microbiome at the tropical cloud forest

FEAST analysis revealed that the organic matter deposited in the bromeliad tanks contributed, on average, 30% to the microbial composition of the fly larvae (Fig. 4). However, there was considerable variation between the fly taxa, particularly in Austrophorocera, which exhibited a coefficient of variation of 73.8%. In contrast, Copestylum showed a lower coefficient of variation (33.2%). Bromeliad tanks shared only six ASVs with Austrophorocera, but they shared 58 ASVs with Copestylum (Fig. 4). Overall, when considering all three species, seven ASVs were shared between the bromeliads and the two fly taxa. Interestingly, the two fly taxa shared only four ASVs between each other, despite both being found in the same organic matter within bromeliad tanks. A complete list of shared taxa among the fly larvae and bromeliad tanks can be found in Table S2.

Figure 4 Contributions of epiphytic bromeliad tanks to the microbiome of two fly larvae.

Left figure shows the proportion accounted by bromeliads as a source of microbes for each individual of Austrophorocera sp. and Copestylum sp. flies. Right figure shows a Venn diagram with the number of shared ASVs between bromeliad tanks and the flies (overlap areas in light green).

Discussion

To gain deeper insights into the complex interplay of factors sustaining the symbiotic relationships between epiphytic bromeliads and canopy insect species in tropical montane cloud forests, we performed microbial community profiling to assess the relative contribution from T. multicaulis tank microbiome to the Austrophorocera and Copestylum microbiome. Our analysis revealed distinct microbial compositions between the bromeliad tanks and the fly larvae. Moreover, we found that the tank of bromeliads significantly contributed to shaping the fly larval microbiome. Approximately 30% of the fly larval microbial communities were tracked to the organic matter deposited in the bromeliad tanks, raising the possibility of microbial exchange between these phytotelmata components and the Diptera larvae. By leveraging microbial community profiling, our study provides valuable insights into the complex symbiotic relationships between species, contributing to a deeper understanding of the ecology and biodiversity of tropical cloud forests and their microhabitats.

Bromeliad tanks serve as ideal microhabitats that support a wide array of microorganisms. The water-filled tanks of bromeliads contain stagnant water, decaying plant and animal matter, and soil, providing nutrients that sustain diverse microbial communities (Carrias, Cussac & Corbara, 2001). Our characterization of the microbiome in the organic matter of T. multicaulis tanks revealed that the microbial assemblages were predominantly composed of bacteria from the phyla Acidobacteriota, Pseudomonadota, and Verrucomicrobiota. Similar composition has been recorded in terrestrial tank bromeliads from a Brazilian sand dune forest (Louca et al., 2017). Moreover, these bacterial phyla have been reported as common inhabitants of acidic (pH ~ 5.1) soils (Spyridonov et al., 2021; Zverev et al., 2023; Borsodi et al., 2024), which is consistent with our results, since the tank samples were collected from the organic matter deposited in the tanks. The dominance of these bacterial groups suggests they play a crucial role in the decomposition of organic matter, and in both carbon and nitrogen cycling occurring within the bromeliad tank ecosystem (Goffredi, Kantor & Woodside, 2011; Lladó, López-Mondéjar & Baldrian, 2017; de Jonge et al., 2023). These findings highlight the importance of bromeliad tanks as self-contained, nutrient-rich microhabitats that support complex detritus-based trophic webs and ecosystem processes in which microbes are involved.

Bromeliad tanks can act as an environmental source of microbes for fly larvae. This is supported by the proportion of the fly microbiome tracked to the microbial communities from the organic matter of bromeliad tanks. Previous studies have suggested that each bromeliad tank contains a unique pool of microbes associated with specific microhabitat conditions, such as pH, and carbon and nitrogen content (Brandt, Martinson & Conrad, 2017), which could lead to varying levels of microbial transfer to a host (Bright & Bulgheresi, 2010). However, our findings indicate low microbial within-community dispersion and consistent microbial profiles among T. multicaulis bromeliads, suggesting that microbial communities were very similar across tanks. Thus, the availability of microbes capable of inoculating the two fly taxa was consistent across all sampled bromeliad tanks. This agrees with other findings in vertebrate and invertebrate species in which the environment is an important filter limiting microbial colonization (Kikuchi, Hosokawa & Fukatsu, 2007; Bright & Bulgheresi, 2010; Loudon et al., 2016; Skelton et al., 2017).

Despite inhabiting the same microhabitat, the fly larvae exhibited distinct microbial communities and varying proportions of shared microbes with bromeliad tanks. In this context, Copestylum larvae had higher microbial diversity and shared a larger number of taxa with bromeliads compared to Austrophorocera larvae. The contribution of bromeliad tanks to the fly microbiome was more consistent for Copestylum than for Austrophorocera larvae, as the latter showed greater variability in the proportion of microbes traced to the bromeliad tanks. This suggests that the environment alone does not fully explain the differences in the fly microbiomes. Instead, the observed variation in the microbiome between fly taxa could be related to species-specific factors, such as species-level differences in physiology, immune function, diet, and functional traits (Buchon, Silverman & Cherry, 2014; Douglas, 2018; Srivastava et al., 2023), which allow for distinctive levels of microbial colonization within a single microhabitat.

Bromeliad phytotelma and the decaying matter provide substrates that may support the microbiome of Diptera taxa. The microbiome of Austrophorocera larvae was dominated by bacterial families commonly found in soil, including Beijerinckiaceae, Microbacteriaceae, Desulfovibrionaceae, and Rhizobiales. These soil-dwelling bacteria likely play important roles for the environment, since they are well-known for their ability to fix atmospheric nitrogen (Morawe et al., 2017; Lindström & Mousavi, 2020; Zhang et al., 2022). The presence of nitrogen-fixing bacteria in the Austrophorocera microbiome suggests they may significantly contribute to the larval nitrogen budget. Another notable component of the Austrophorocera larval microbiome was the family Chitinophagaceae; these bacteria are specialized degraders of chitin and cellulose, two of the most abundant biopolymers in nature (Wieczorek, Hetz & Kolb, 2014; Li et al., 2016). The high relative abundance of Chitinophagaceae in Austrophorocera larvae indicates they likely play a role in breaking down these complex carbohydrates available in the bromeliad tanks, potentially aiding in the digestion of fungal and plant material consumed by the larvae.

Copestylum larvae showed higher microbial diversity compared to Austrophorocera larvae. These larvae are saprophagous, meaning they can feed on water-borne or firmer decaying organic matter (Rotheray, Hancock & Marcos-García, 2007). A previous study examined the gut microbiome composition of different Copestylum species, revealing a large proportion of bacteria belonging to the Actinomycetota and Pseudomonadota phyla (specifically, Alphaproteobacteria and Gammaproteobacteria) for C. latum, and Enterobacteraceae for C. limbipenne (Martínez-Falcón et al., 2011). Our findings partially agree with this previous research. The microbiome of Copestylum larvae from bromeliad tanks was mainly dominated by bacteria belonging to Chitinophagaceae (Bacteroidota), Beijerinckiaceae (Pseudomonadota), Bacteroidaceae (Bacteroidota), and Clostridiales (Bacillota). These bacteria have been associated with the degradation of organic carbon commonly found in sediments (Yu et al., 2023). The over-representation of these bacteria in the Copestylum larvae microbiome suggests that they could be providing benefits to the host through carbon metabolism, potentially aiding in the digestion and utilization of the decaying matter present in the bromeliad phytotelma. Overall, the microbial profiles found in these two fly taxa seem to be related to the life history of these insects.

The tropical montane cloud forest is one of the most diverse yet also one of the most threatened biomes in the world (Scatena et al., 2010; Karger et al., 2021; Ramírez-Barahona et al., 2025). While the role of tank bromeliads in supporting canopy fauna and providing ecosystem services in cloud forests has been well documented (Ladino et al., 2019), little is known about their specific role in mediating the microbiome acquisition of the insect species they harbor. To address this knowledge gap, our study investigated the complex ecological relationships that occur in disturbed tropical cloud forests, focusing on the interactions between epiphytic vascular plants and fly larvae that inhabit bromeliad phytotelmata. We showed that bromeliad tanks may be a source of microbes for fly larvae that utilize the organic matter deposited in the tanks during a critical developmental stage in their life cycle. However, the amount of microbial transfer varies depending on the host species. This finding enhances our understanding of the intricate web of interactions supported by these epiphytic tank bromeliads in tropical cloud forests.

Conclusions

We conducted a study to investigate the role of epiphytic tank bromeliads as environmental sources of microbes for two fly taxa. Our findings reveal that the organic matter within the tank of the epiphytic bromeliad T. multicaulis significantly influences the microbiome of fly larvae. The organic matter collected in bromeliad tanks creates distinctive microhabitats in the forest canopy, where various nutrient cycling processes occur. These unique conditions likely play a crucial role in shaping the microbiome of fly larvae. Interestingly, the extent to which these microhabitats contribute to the composition of the fly microbiome varies, suggesting that additional factors, such as genetics and diet, also influence microbial diversity. To further our understanding, future research should investigate whether these microbial compositions persist into the adult stage of the flies. This exploration could provide valuable insights into the ecological interactions between bromeliads and their associated fauna.

Supplemental Information

Supplemental Information 1 Microbial distances to the cluster centroid to assess within-community dispersion.

(A) Data based on unweighted UniFrac distances. (B) Data based on weighted UniFrac distances.

Supplemental Information 2 Enriched microbial taxa in bromeliad tanks and in two Diptera taxa.

Supplemental Information 3 Shared microbial taxa between bromeliads and fly larvae of two distinct taxa.

Supplemental Information 4 Scripts and working files for microbial analyses.

We thank Amelly Hyldaí Ramos Díaz and Luis Argel Delgado Vásquez for their assistance in the field.

Additional Information and Declarations

Competing Interests

The authors declare that they have no competing interests.

Author Contributions

Rodolfo Martínez-Mota conceived and designed the experiments, performed the experiments, analyzed the data, prepared figures and/or tables, authored or reviewed drafts of the article, and approved the final draft.

Antonio Acini Vásquez-Aguilar performed the experiments, authored or reviewed drafts of the article, and approved the final draft.

Dolores Hernández-Rodríguez performed the experiments, authored or reviewed drafts of the article, and approved the final draft.

Emilio A. Suárez-Domínguez conceived and designed the experiments, performed the experiments, authored or reviewed drafts of the article, and approved the final draft.

Thorsten Krömer conceived and designed the experiments, performed the experiments, authored or reviewed drafts of the article, and approved the final draft.

DNA Deposition

The following information was supplied regarding the deposition of DNA sequences:

The sequences are available in the NCBI Sequence Read Archive: PRJNA1146765.

Data Availability

The following information was supplied regarding data availability:

The data and code are available at Dryad: Martínez-Mota, Rodolfo (2025). Data for analyzing the microbiome of bromeliads and fly larvae inhabiting the tropical cloud forest of Mexico [Dataset]. Dryad. https://doi.org/10.5061/dryad.6wwpzgn78.

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
