# Peer review of "Close neighbors, not intruders: investigating the role of tank bromeliads in shaping faunal microbiomes"

_PeerJ, doi:10.7717/peerj.19376_

## Round 0.1 · original submission · Major Revisions

Dear Dr. Martínez-Mota,

Your paper has been reviewed by two experts in the field. They agree that your research was well-executed, and your manuscript provides relevant information on the interaction between bromeliads' microbiota and fly larvae. However, they also provided important suggestions that I hope you address. After you revise the manuscript following the reviewer's suggestions, I will be pleased to reconsider the manuscript for publication in PeerJ. Please make sure to acknowledge the reviewer's valuable contributions to the revised version.

Reviewer 1 ·

Basic reporting

no comment

Experimental design

no comment

Validity of the findings

no comment

Additional comments

Dear editor and authors,
Here, I provide my review of the manuscript entitled “Close Neighbors, Not Intruders: Investigating the Role of Tank Bromeliads in Shaping Faunal Microbiomes.” Overall, it is a well-conceived study; the manuscript is well-written and clear, and the analyses effectively address the general aim of the research. The figures are also very helpful. I have only minor comments / suggestions, which I have included directly in the .docx file
PS: Please disregard comment A3 in the PDF file.

Annotated reviews are not available for download in order to protect the identity of reviewers who chose to remain anonymous.

·

Basic reporting

1. Usage of the English language is adequate. I just have a few minor flow/grammar points listed below.
Line 58: fulfill
Line 59: , and forest nutrient and water cycling
Lines 61 - 66: the sentence is too long, making it hard to follow. Please divide it in two sentences
Line 68: “still remains” instead of “is still”
Line 69 – 70: such a strong statement should be supported by more references
Line 90: decaying organic matter
Line 108: units?
Line 141: replace “, thus,” by “. Therefore,” or similar
Line 151: Mention here that you are focusing on a single species, otherwise it sounds like you are considering different species within each genus as a single unit of analysis
Lines 155 – 166: I do not understand what you mean by “differences in microbial communities between variables”. Which variables are you referring to? Please rephrase.
Lines 186: an increase along what? It sounds like you are comparing the two groups along some kind of axes, but maybe you just mean that the absolute values are higher in one species than the other?
Lines 222 – 223: commenting results is usually best done in the discussion. Remove “which are typically found in wet soils”
Lines 244 – 245: if you want to keep “the” before “tropical montane cloud forest”, add the geographic location of the specific forest, or remove “the” and make forest plural
Line 248: Why however? What follows does not seem to go against what is said in the previous sentence
Line 258: I am not sure than “slurry” is an adequate word here. I understand slurry as “suspended solids in water”, but, if undisturbed, most organic matter in bromeliads usually falls and accumulates at the bottom of the tank. It is in this benthos, that are found most bromeliad organisms, including those examined in the study. If what was actually sampled for bacteria was the water column, then it would not be the exact habitat in which the studied larvae are found.
Line 264: Here as substrate, I understand that you mean the solid organic matter within the tank, but it can also be understood as the substrate of the bromeliad itself. Maybe remove the word to avoid any ambiguity
Line 264 – 266: this sentence feels a bit out of place, either modify or blend with the preceding sentences
Line 307: “withdrawn” does not seem very appropriate here, maybe you meant “collected”?
Line 327: “symbiotic relationships among insect fauna.” sounds like there are symbiotic relationships between the different insect species, which is not necessarily reflected in the literature, and not the point of the paper. Rephrase with “mediating the microbiome acquisition of the insect species they harbor” or similar
Lines 329 – 330: strictly speaking, these larvae live in the bromeliads rather than benefit from them
Line 641: UniFrac is capitalised in the main text, but not in the figure captions

2. Relevant literature is cited throughout the manuscript, although I would add the following references to improve contextualisation.
To metamicrobiome concept, when discussing that relating the microbiome in the environments and that of animals in the context of decomposition:
de Jonge, I. K., M. P. Veldhuis, J. H. C. Cornelissen, M. P. Berg, and H. Olff. 2023. The metamicrobiome: Key determinant of the homeostasis of nutrient recycling. Trends in Ecology & Evolution 38: 183–195.
To more adequately contextualise differences between the two different species, additional information on their functional group and/or traits can be added to make the assertion round “host-specific factors” more precise:
Céréghino, R. et al. 2018. Constraints on the functional trait space of aquatic invertebrates in bromeliads. Funct Ecol 32: 2435–2447.
Srivastava, D. S. et al. 2023. Geographical variation in the trait‐based assembly patterns of multitrophic invertebrate communities. Functional Ecology 37: 73–86.

3. I think the author should change the text lines 73 – 79. Given that the authors discuss the potential for bromeliads to be a source for the microbiome of local fauna, it feels a bit out of place to discuss ecological disturbance and degraded ecosystems. These would lead the reader to wrongly think that they are central themes of the paper.

4. The authors would also need to mention why they chose to study these two fly taxa in their study. Was it because they are abundant? Or because they represent a special function and thus can yield unique insights in the question at hand?

5. Figures are high quality. However, it seems unclear why the same colours are referred to with different names in the different captions, e.g. violet vs. dark orchid. In addition, in Figure 4, the colour representing bromeliads is also green, so maybe write “light green” for areas of overlap.

Experimental design

6. In Material & Methods, I would stick to either the active or the passive voice, as repeated switch between the two tend to break the flow of the paragraphs.

7. Why did you pick these two genera? Please give at least the taxonomic family to the main text so that naïve readers can quickly understand which kind of organism is discussed.

8. Lines 149 – 150: Please briefly mention why you used ASV and Faith’s distance, similarly to what you did later in the paragraph.

9. It seems unclear to me why the authors decided to plot betadisper results with an NMDS, while the betadisper is conceptually closer to a Principal Coordinate Analysis (PCoA) or a Principal Component Analysis (PCA). Moreover, NMDS should be reported with stress values, to assess if they adequately represent the data.

Validity of the findings

10. A very important finding of the paper is that there is a lot more variability in Austrophorocera than in Copestylum. However, when the discussion reaches a description of these two microbiomes, little mention is made of this result. Even in the paragraph focussing on Austrophorocera, the wording is such that it seems that there is little variation among the individuals, with some constant patterns.

11. The conclusion seems very brief considering the very interesting results of the manuscript. It would deserve a sentence or two with details on the results and discussion.

Additional comments

12. The provided script does not load all the required data for the analysis, e.g. bromeliads_df

13. I would highly recommend that the supplied data and analysis folder follows the rules of open reproducible science. Right now, the supplied folder has everything needed to reproduce the study, but the current organisation make it hard to use. To give just one example, paths in the R script do not include the relative paths to where the relevant data files are, so the user has to enter them by hand.
Here are several examples of patterns that you could follow.
https://guides.lib.virginia.edu/RDM/file-management
https://www.earthdatascience.org/courses/intro-to-earth-data-science/open-reproducible-science/get-started-open-reproducible-science/best-practices-for-organizing-open-reproducible-science/


This would also help quickly understanding the structure of the data, more than just describing what the data is for in the naming system. To give an example, I am not used to microbial analysis, and it took me a while to figure out that the raw composition data was in the .qza files.

---

## Round 0.2 · Major Revisions

Dear Dr. Martínez-Mota,

Your paper has been reviewed by two experts in the field, who still raise some important points that should be addressed before its publication. Please, I kindly ask you to address the reviewer's suggestions. Then, I will be pleased to reconsider the manuscript for publication in PeerJ.

·

Basic reporting

1. Basic reporting
1. I appreciate the authors incorporating comments from the last review. I just have a few minor additional points.

Line 24: Using “The” makes it sound like the authors are talking about a specific forest, but the second half of the sentence makes it sound like it is general. Maybe the authors can remove “The” and make it plural, or give a specific location.

Line 29: Here it sounds like the authors look at all fly larvae, while they focus on two species.

Line 33: The families should be mentioned here too

Line 44: “larvae, “ should be removed here, otherwise it is mentioned twice in the same sentence that the focus is larvae

Line 54: “these forests” is plural so “its” should be “theirs”. Even better, replace by “the plant species of these systems” because it is hard to find what the pronoun refers to

Lines 60 - 62: This sentence repeats the sentence above, maybe the authors can remove the second half and blend with sentence below? Alternatively, they could emphasise the rosette more as the mechanism behind the services mentioned in the previous sentence

Line 65: serves

Line 67: here could be a good place to have a sentence or two on the research done on microbes within bromeliads (e.g. Gonçalves et al. (2014), Louca et al. (2017), Rogy and Srivastava (2023), between others), i.e. to emphasise the knowledge gap the authors fill with their study

Line 106: please include the genera and families of the larvae here too. Keep it in line 183 so that the reader has a reminder

Line 107: obtain sufficient replicates to have high confidence in our results, or similar

Lines 168 – 169: It is good practice to cite packages, but maybe the citation can be moved right after the package name so that it is easy to know which refers to which.

Line 223: belonging to the WD2101

Line 311 – 312: this sentence makes it sound like the authors collected many different fly taxa

Experimental design

2. My biggest comment has to do with the assertion that bacterial taxa that are shared between the bromeliads and the larvae has to come from the bromeliads. If I understood the design well, the authors collected the bacteria both in the bromeliads and in the larvae these contained. The study is thus observational and not manipulative. In this case, how can we be sure that the bromeliad is the source of the shared taxa? Arguably, this can only be achieved by growing/collecting the larvae in different environments and then comparing their microbiota. The study takes a different approach, where the source is previously defined in the FEAST analysis. This is totally fine, as it is akin to causal analysis, but my concern is that because only three variables were measured (the bromeliads, and two groups of larvae) there are not enough elements to assign the variation to the bromeliads themselves. For this comment, the authors can either tone down the causality language, or develop why their method can give them enough confidence to have this causal link.

3. Line 116: I am a bit confused here, it sounded like the focus was all bacteria (body surface + gut), but now it sounds like body surface bacteria play little to no role in the results? Could you clarify that? If body surface contributes little to the result, then maybe it is not worth emphasising repeatedly.

Validity of the findings

4. “A very important finding of the paper is that there is a lot more variability in Austrophorocera than in Copestylum. However, when the discussion reaches a description of these two microbiomes, little mention is made of this result. Even in the paragraph focussing on Austrophorocera, the wording is such that it seems that there is little variation among the individuals, with some constant patterns. “ Although the authors have incorporated part of this comment in their new version of the manuscript, this result deserves a bit more discussion. For example, why discussing the diet of Copestylum only? This discussion needs not to be extensive, but the authors should have an educated, grounded guess at potential mechanisms behind the observed differences. If space is the issue, the last paragraph of the discussion is essentially the same as the conclusion, so could be replaced by a short discussion on potential mechanisms.

Additional comments

I appreciate all the work that the authors have put in improving the data and code. However, I still have comments relating to the reproducibility of the code.

5. The current folder system still does not allow me to easily reproduce the study. First, the author should use relative and not absolute path, i.e. not force the user to put the folder in the root directory "~/" to be able the run the code. Because I work on window and did not want to put the folder in my root directory, I had to manually change all paths to be able to run the code... The solution here is easy: simply create an R project in the base folder ("bromeliads-scripts-and-working-files/") so that wherever the folder is, the user can just load the project and automatically set their working directory to wherever the folder is on their machine. For help with R projects within Rstudio: https://support.posit.co/hc/en-us/articles/200526207-Using-RStudio-Projects

6. The paths to load the file have not been updated: the working directory is set to "bromeliads-scripts-and-working-files/03-working-files" while the files are in a different folder now. NB: with Rprojects no need to set working directories

7. From line 53 onwards, I am receiving the warning that there is more than one “phylo” class in the cache, and that only the first one will be used. If this is normal, can there be a comment explaining why?

8. The function line 231 about the Venn diagram gives an error and does not run, maybe it is about the warning in the line above?

9. I am not able to run the code for metadata, I get an error line 241. So I have not been able to run the script after this line

10. ggplot2 and dplyr are included in the tidyverse package, so there is no need to call them separately if there is a call library(tidyverse)

11. The .DS_Store files in the different working directories can be removed. These mac-specific files store some computer-specific parameters, and can mess with those of other mac users trying to run the code

12. The .Rdata and .Rhistory files should also be deleted. These store the R working environment and the history of the commands that were used, thus should not be shared with future users.

Reviewer 3 ·

Basic reporting

The manuscript is well-writen and about a relevant topic. My only concern is about the Introduction structure, which does not follow from the more general to the more specific subjects. The authors must start it with the ecological background. The bromeliad and the flies are just study models. I also would like to see some explanation about why these models were chosen.

Experimental design

The manuscript is exploratory, and a substantial number of plants were surveyed (n=17). The reasarch question was well-defined and the manuscript is suitable for publication in PeerJ.

Validity of the findings

The methods are clear and replicable, the data was provided, and the findings are valid.

Additional comments

The authors did a good job addressing the concerns the previous reviewers pointed out. My only point regards the Introduction structure, which can be improved.

---

## Round 0.3 · accepted · Accept

Dear Dr. Martínez-Mota,

Thank you for addressing the reviewer's suggestion. They have not agreed to re-review, but I have assessed the revision myself and am happy with the current version. I believe your work makes a valid contribution to the field and is ready for publication.
Best regards,

Guilherme Corte